# HIV policy legacies, pandemic preparedness and policy effort to address COVID-19

**Ashley Fox** [ID]*, **Heeun Kim** [ID]

Department of Public Administration and Policy, Rockefeller College of Public Affairs and Policy, University at Albany, Albany, NY, United States of America

* afox3@albany.edu

## Abstract

That the world was unprepared for a major infectious disease outbreak is now readily apparent to all credible observers. However, some countries were more prepared than others and we have seen a variety of responses to COVID-19 emerge across nations. While recognizing that the sources of variation in country responses to COVID-19 are many and varied, in this study we seek to examine how policy legacies from national responses to HIV have influenced countries' responses to COVID-19. The aim of this study was to examine whether countries with a more conducive HIV policy environment were better prepared for COVID-19 and have therefore had more preemptive and rights-based responses. Using data from the Oxford Covid-19 Government Response Tracker, we develop measures of country effort to respond to COVID-19 including early containment and closure policies, prevention policies, economic policies, and health system policies. We combine this with data from the HIV Policy Lab and correlate overall and disaggregated country HIV Policy scores with COVID-19 Policy scores. We find that the COVID-19 Containment and Closure Measures Index was negatively correlated with supportive social policies related to HIV in the early stages of the pandemic, but the association did not persist as time went on. The COVID-19 Economic Support Measures had prolonged positive associations with supportive social policies related to HIV and negative association with clinical and treatment policies. Countries with stronger structural responses to HIV have been less inclined towards involuntary measures and more prepared for the social and economic elements of COVID-19 pandemic response.

## Introduction

The COVID-19 outbreak has been a test of global health pandemic preparedness. By most accounts, countries failed to respond in an efficacious manner that both drove down cases to sustainable levels while minimizing harms in other areas. To control spread and reduce mortality, countries have adopted a range of policy responses including various forms of "lockdown" measures (e.g., school/workplace closings, cancelling large events, restrictions on gatherings/movement outside the home), travel restrictions (internal and external), economic policy responses (e.g., income support, debt relief), and prevention policies (e.g., face coverings, contact tracing, testing, investment in vaccines). Some countries adopted effective strategies to

**Data Availability Statement:** The replication materials are available at Harvard Dataverse: https://doi.org/10.7910/DVN/R3HZGV.

**Funding:** The authors received no specific funding for this work.

**Competing interests:** The authors have declared that no competing interests exist.

reduce transmission more rapidly than others. To prepare for future pandemics, or even to inform current policy towards managing and living with COVID-19, it is important to understand the factors that influenced countries' adoption of countermeasures against COVID-19. While recognizing that the sources of variation in country responses to COVID-19 are many and varied, in this study we seek to examine how policy legacies from national responses to HIV have influenced countries' responses to COVID-19. We do so by examining the association between countries HIV Policy Index score from the HIV Policy Lab with their Oxford COVID-19 Government Response Tracker (OxCGRT) scores. We discuss implications for pandemic preparedness and policy learning during acute emergencies.

## Background

Although there have been several recent global infectious disease outbreaks (e.g., SARS, H1N1; Ebola, Zika), in some respects, the most comparable recent infectious disease pandemic to COVID-19 is HIV in terms of the breadth of its spread and the magnitude of the response. Whereas other outbreaks have remained more geographically contained, HIV spread rapidly around the globe and has been identified in every country around the world, much like COVID-19. HIV emerged at a time when infectious diseases in high-income countries appeared to have been vanquished and has received high degrees of worldwide attention and resources [1, 2]. Countries have set up infrastructure to secure their blood supply, built National AIDS Commissions, developed HIV sero-surveillance systems, and adopted policies aimed at prevention, treatment, care and support [2–4]. HIV is the only disease with its own dedicated United Nations agency. The global investment in a search for a suitable vaccine as well as the political barriers to ensuring affordable treatment access for all share many parallels with current efforts to scale up COVID-19 vaccine access globally [5].

Researchers have already begun to transfer lessons from the global fight against HIV to the COVID-19 response [6–12]. Below we summarize some of the lessons and infrastructure from national HIV policy responses that may theoretically be valuable to policy learning and improved COVID-19 responses, especially in low- and middle-income countries (LMICs).

First, from the clinical infrastructure perspective, in response to HIV and other infectious disease outbreaks, including Ebola, SARs and MERs, countries have had to build their clinical and testing capacities [13]. This has provided critical infrastructure to strengthen their responses to other emerging infectious diseases [14]. For instance, Hasan et al. (2022) note that the surveillance network built for HIV in Africa can used towards the COVID-19 response [15]. Studies have also noted individual case studies highlighting how investments from HIV and recent experiences with Ebola have contributed to the speed and organization of its COVID-19 responses, since officials could use already established and well-used coordination structures. For example, in Senegal, the presence of the Institut Pasteur Dakar, a major infectious disease research institute, was instrumental in scaling up testing capabilities. The Institute was one of two labs in Africa able to test for COVID-19 after initial cases were detected, and its experts subsequently trained many other labs across the continent [16, 17]. Senegal also built on its recent experience with the West African Ebola epidemic during which it had developed a Public Health Emergency Operations Center and began running emergency drills and stockpiling personal protective equipment [16].

Likewise, national COVID-19 responses can build on capacity building efforts and surveillance and monitoring systems that have been developed over many years of investment by national authorities and The Global Fund in relation to tuberculosis control in addition to HIV. As an opportunistic infection, tuberculosis experienced a resurgence in heavily AIDS-affected countries and populations. Tools such as GeneXpert, chest radiography and case-

finding strategies can be applied to the detection and treatment cascade in relation to COVID-19 [18]. For instance, Nigeria's PEPFAR program supported the scale-up of a mobile diagnostic facility called "Wellness on Wheels'" (WOW Truck). The truck was equipped with a mobile GeneXpert system that had the capacity to test for TB, HIV, and COVID-19 in hard-to-reach communities, which significantly reduced the turnaround time for test results [19].

Additionally, in the absence of effective treatments and prophylactics, early HIV interventions shared with COVID-19 the necessity of relying on more classical social and behavioral interventions [10, 20]. Therefore, countries with more mature HIV programs may have had a better trained and prepared public health workforce and set of civil society actors to draw on to implement behavioral interventions and communication strategies. In response to the need to address HIV and extensive donor funding in this area, countries have also trained infectious disease experts who have spent their careers on HIV who could quickly pivot and apply their expertise to COVID-19 [21]. Thus, existing infrastructure and capacity building efforts related to HIV may have enabled a more proactive and appropriate COVID-19 response.

Second, early in the HIV pandemic response, global HIV advocates recognized the need to build health system capacity to respond effectively to HIV [22]. Global attention to health systems strengthening largely grew out of the necessity of HIV programs to deliver care and treatment to people living with HIV [23, 24]. Community health-worker programs were scaled up and attention to building human resources for health grew substantially [25, 26]. Studies have noted the intersections of scaling up global health security through building primary health care capacity and universal coverage including a Lancet Commission on synergies between UHC, health security, and health promotion [27].

Third, critical lessons can be gleaned from the health and human rights and advocacy movements that developed in response to the HIV pandemic. Jonathan Mann, former head of the World Health Organization's Global Program on AIDS, was a pioneer in defining human rights as the core of the public health response in his work responding to the emerging HIV epidemic in Zaire (Democratic Republic of Congo) [28]. Mann insisted that the promotion and protection of health is inextricably tied to the promotion and protection of human rights and dignity [28]. Mann rejected the notion public health measures should be used as justification to violate human rights.

The health and human rights turn in public health heavily influenced approaches to HIV management and prevention by prioritizing protecting patient confidentiality, engaging affected communities in responses and reducing stigma by rejecting discriminatory policies towards most at risk groups and criminalization of transmission. The Siracusa Principles grew out of this movement, which suggest that public health measures should use the 'least restrictive means possible' to achieve public health objectives and should be subject to review against abusive applications [29, 30]. Countries with more experience with implementing 'rights-based' approaches may be more attentive to protecting human rights during COVID-19 and avoiding overly coercive strategies that fall disproportionately on marginalized groups.

Relatedly, research on gender and HIV transmission highlighted the importance and complementarity of economic policy responses to counter structural drivers of HIV transmission [31, 32]. Behavior change intervention that fail to appreciate how material conditions contribute to 'risk behaviors' were shown to be ineffectual [33, 34]. Likewise, experience with treatment scale-up exposed adherence to medication requires not only information and motivation but also addressing food insecurity and other material obstacles to appropriate use [35, 36]. Thus, countries with more experience implementing structural interventions may be more aware of the need to offset the economic burdens of public health requirements to compel compliance.

Of course, while all infectious disease emergencies share some similarities, each disease varies in important ways in terms of the dynamics of transmission, the severity of illness, the availability of treatment, and evidence-based control measures that render past experiences less comparable. For instance, whereas quarantine measures were sharply rejected in the case of HIV where transmission requires intimate contact, the specific disease profile of COVID-19 with its high transmissibility via droplets and aerosols makes it a better candidate for quarantine measures. Similarly, the need to rapidly scale up hospital bed and acute care capacity while isolating infected patients is a much larger priority for COVID-19 than HIV. Here, experiences with Ebola were more informative in demonstrating the problem that limits to hospital capacity can contribute to mortality in other areas as the health system becomes overburdened. Nevertheless, recognizing these differences, the purpose of this study is to examine how the nearly two-decade investments in HIV prevention and treatment, and countries differential capacity in this area, may be associated with their COVID-19 policy responses.

Previous research has shown that past disease responses in countries condition future disease responses. For instance, Baldwin (2005) finds that past responses to contagious diseases conditioned countries' responses to HIV, leading some European countries to adopt harsher rights-constraining policies rather than more moderate, rights-enabling policies [2]. Likewise, Nunn (2009) finds that institutions developed in the early stages of Brazil's HIV response influenced subsequent aspects of its response and conditioned government commitment to responding to HIV [37]. Robinson (2019) finds that because HIV and unwanted pregnancy share heterosexual sex as the primary causative mechanism, historical variations in family planning provision offer insight into the relative success of HIV prevention during the first decades of its spread [38]. Since family planning initiatives and HIV prevention programs both ultimately seek to regulate the same sensitive behaviors, nations with family planning policies and NGOs established prior to the onset of the HIV epidemic were best primed to share the resources, discourses and strategies which had been developed to limit unwanted pregnancy. National and international HIV prevention programs found that they were able to utilize and modify the systems and donor bases, which had grown up after years of hard-won progress in family planning.

The aim of this study was to examine whether countries that exhibited a greater degree of commitment to fighting HIV were better prepared for COVID-19 and have therefore had more effective and appropriate responses. This may occur through at least two different mechanisms: 1. *State Capacity*. Countries with recent pandemic experience have built capacity and skills to respond to future pandemics such as through the development of surveillance systems, laboratories, testing infrastructure, the development of contact tracing protocols, and training of a public health workforce, infectious disease experts and civil society organizations that can be mobilized and temporarily shifted to the COVID-19 response. 2. *Social policy learning*. Countries learn from past outbreak responses both positively and negatively about what worked and did not work. This learning includes elements about the social aspects of disease responses. For instance, HIV taught the global health community important lessons about the need to consider stigma, marginalization, socio-economic factors, risk communication and the necessity of catalyzing public attention to raise the saliency of health threats in the face of public complacency [6, 7, 39]. Whether this learning has transferred to produce a more "humane" or rights-based approach to addressing COVID-19 can also be assessed [40].

It is presently unclear how effectively these policy lessons have been transferred to the COVID-19 outbreak. For instance, most countries were quite slow in scaling up their testing capacity, even though testing is known to be one of the few effective preventive tools in an infectious disease outbreak in the absence of effective treatments and vaccines [41]. Most countries still had inadequate testing capacity even a year and a half into the pandemic.

Likewise, up until the COVID-19 pandemic, public health had largely moved away from draconian measures to combat infectious disease threats, preferring instead "rights-based" approaches that sought to balance the need to protect public health from disease threats against the harms that might arise from the disease response itself with the goal of finding the "least intrusive means possible" to achieve its purposes [39]. Longstanding precedent, such as not closing borders, were rapidly reversed in favor of nationalistic responses. Even right up to the first case appearing in the US, experts questioned the feasibility of mass quarantine in advanced-industrial democracies while recognizing that it was certainly possible [42]. Undertaking restrictive measures early on as a means of limiting initial spread may be justified to prevent more restrictive measures over a longer term, but many of these early measures proved longstanding as countries cycled in and out of lock-down measures in response to rising and falling cases and never seemed to rely primarily on less restrictive approaches. More research remains to be done to determine whether "stricter" lock-down measures were "better," [43] or in other words whether the benefits of strict stay-at-home orders were justified when balancing their contribution to a reduction in cases of COVID-19 against the additional harms that these policies engendered, especially in LMICs with more favorable demographic profiles and greater economic insecurity.

A simple association between a countries' overall HIV policy response and their COVID-19 policy response may therefore mask additional nuances associated with policy learning. For instance, less stringent measures may be preferred by countries that have learned from their HIV responses to protect human rights and prefer more voluntary measures. Countries that were better prepared in terms of their clinical and treatment regimens for HIV might be better prepared to roll out testing, contact tracing and vaccination. Countries that have adopted supportive social policies related to HIV may be more likely to recognize that coercive policies cannot succeed without providing economic support to the public. To tease apart whether different mechanisms may come into play for sub-components of policy responses, we therefore also examine associations between sub-measures of policy responses. Our overall hypotheses and sub-hypotheses are as follows:

*H1. Exhibiting a higher HIV Policy Index Score will be associated with a higher COVID-19*

 *Policy Measures Index Score reflecting more aggressive measures undertaken.*

 Sub-hypotheses:

*H2A. These associations should be strongest after the initial 30 days of a country's COVID-19 Policy Measures Index Score when a more nuanced policy response could have had time to develop.*

*H2B. Countries exhibiting a higher HIV Policy Index Score on "Structural Determinants" (e.g., protection of human rights) will be associated a lower COVID-19 Containment and Closure Measures but a higher COVID-19 Health Systems Measures and Economic Support Measures. This will reflect policy learning from HIV about the importance of 'rights-based' approaches and addressing structural drivers.*

## Methods

### Data

To assess the association between the HIV policy responses and COVID-19 policy responses, we use data from HIV Policy Index [44] and the Oxford COVID-19 Government Response Tracker (OxCGRT) [45].

**HIV policy index.** The HIV Policy Lab is a collaboration among academic institutions, the UN, and civil society organizations, which tracks and measure HIV-related laws and policies across 194 countries around the world [46]. The HIV Policy Index consists of 33 measures across four major categories: 1. Clinical and Treatment; 2. Testing and Prevention; 3. Structural; 4. Health Systems. The detailed description of each measure was presented in S1 Table.

For each policy category and overall, each country receives a HIV Policy Lab summary score. For all indicators for which there are data, the total of adopted (= 1) and partially adopted (= 0.5) is divided by the total number of indicators scored, then multiplied by 100 to create the continuous score. In coding different policies, the focus is on the content of the law and policy in a country and not on whether or to what degree that policy has been implemented. Only countries that had data collected on at least one third of all indicators for the policy category and for the overall score are included and where data were missing, the numerator was reduced so that we are only scoring a country based on existing data. Data are available from 2017–2020. We use the average across all time periods available in order to capture the general essence of the HIV Policy Index across the time frame. Table 1 summarizes all measures included in the HIV Policy Index.

**Oxford COVID-19 Government Response Tracker (OxCGRT).** The OxCGRT provides systematic cross-national, time-series data to understand how government responses have evolved over the full period of the disease's spread. The data is collected from publicly available sources such as news articles and government press releases and briefings via internet searches by a team of over one hundred Oxford University students and staff. The OxCGRT captures 23 discrete policies measured across three major categories: Containment and Closure, Economic Support, and Health Systems. Table 2 describes the policies of these three categories. The OxCGRT researchers have created several composite indices out of these measures, i.e., Stringency Index, Containment & Health Systems Index, and Government Response Index, that based on selected policies across the three major categories (Containment and Closure, Economic Support, and Health Systems). Thus, we created another set of composite indicators for this analysis that presents index scores of each three major category. The indices were created by taking the score from the ordinal value of each measure and rescaling each by their maximum value to create a score between 0 and 100. An extra half-point was subtracted from each ordinal score if the policy targeted rather than general. We followed the assumptions and calculations used by the original OxCGRT indices, which treated a missing value as 0 to avoid the risk of inflating the policy effects from little information [45]. Finally, these scores are then averaged to get three composite indices. The data for the OxCGRT spans the period January 2020 to December 2021 (at the time of this study). Below we describe how we treat time in the analysis.

**Controls.** We control for several immediate potential confounders including a countries' universal health coverage index score, HIV prevalence in year 2019, logged GDP per capita in year 2019, and level of democracy in 2019. The Universal Health Coverage Index comes from World Health Organization and measures the average coverage of essential health services such as reproductive, maternal, newborn and child health, infectious diseases, non-communicable diseases, and service capacity and access [47]. HIV prevalence represents a country's HIV prevalence in 2019 and came from the Global Burden of Disease 2019 study [48]. Data on logged GDP per capita came from the World Bank World Development Indicators [49]. We control for countries' level of political democracy with the Polity IV data [50]. The Polity Score captures this regime authority spectrum on a 21-pont scale ranging from– 10 (hereditary monarchy) to + 10 (consolidated democracy) and measures degree of democratic governance along six component measures.

**Table 1. Summary of HIV policy response measures.**

| | |
|---|---|
| **Clinical & Treatment (CT)** | |
| CT1—Treatment initiation | Are all people living with HIV, regardless of CD4 count, eligible to start HIV treatment in national policy? |
| CT2—Same-day treatment start | Is the option to start treatment the same day as HIV diagnosis included in national policy? |
| CT3—Treatment regimen | Are up-to-date first-line ARV regimens aligned with international recommendations included in national HIV policy? |
| CT4—Differentiated service delivery | Do national HIV treatment policies include multiple options for differentiated HIV treatment services? |
| CT5—Viral load testing | Is viral load monitoring at least once per year provided for in national policy? |
| CT6—Pediatric diagnosis & treatment | Are national pediatric testing and treatment policies aligned with international recommendations? |
| CT7—Migrants access to healthcare | Are HIV services and primary healthcare available to all migrants under the same conditions as citizens under national policy? |
| CT8—TB diagnostics | Are rapid diagnostic tests (e.g., rapid molecular diagnostics, TB-LAM) aligned with WHO recommendations used as the initial diagnostic test for TB in PLHIV under national policy? |
| **Health System (HS)** | |
| HS1—Task shifting | Are nurses or other non-physicians allowed to initiate HIV treatment under national policy? |
| HS2—Health financing | Does the national budget and fiscal policy include sufficient health spending and adequate tax revenues to meet international targets? |
| HS3—Universal health coverage | Does national health coverage include medications for HIV treatment & pre-exposure prophylaxis (PrEP)? |
| HS4—User fees | Are public primary healthcare and HIV services available without user fees at the point of service under national policy? |
| HS5—Access to medicines (TRIPS) | Does national law/policy take advantage of TRIPS flexibilities for affordable medicines? |
| HS6—Unique identifiers with data protections | Are unique identifiers for continuity of care across multiple facilities included in national policy along with protections for patients' privacy? |
| HS7—Data sharing | Is it national policy to publicly share disaggregated HIV data on a regular basis? |
| **Structural (S)** | |
| S1—Same-sex sex non-criminalization | Does national law/policy refrain from criminalizing and prosecuting people for consensual same-sex sexual acts? |
| S2—Sex work non-criminalization | Does national law avoid criminalizing sex work (buying, selling, and organizing of sex work)? |
| S3—Drug use non-criminalization | Does national law refrain from criminalizing personal drug use/possession? |
| S4—HIV exposure non-criminalization | Does national law refrain from criminalizing and prosecuting people for HIV exposure/ transmission? |
| S5—Non-discrimination protections | Do national laws/policies include protections from discrimination on the basis of sexual orientation, gender identity, and HIV status? |
| S6—National human rights institutions | Is there an independent national human rights institution to which violations can be reported? |
| S7—Constitutional right to health | Is there an enforceable right to health in the national constitution? |
| S8—Girls education | Is there a national policy in place to encourage secondary school retention among girls? |
| S9—Gender based violence | Does the law explicitly address domestic violence with enforceable penalties? |
| S10—Civil society | Are NGOs/CSOs able to register, seek funding and operate freely under national law and is there a social contracting policy for financing NGOs/CSO-provided services? |
| **Testing and Prevention (TP)** | |
| TP1—Self-testing | Is self-testing approved in national policy? |

*(Continued)*

**Table 1.** (Continued)

| | |
|---|---|
| TP2—Partner notification/Index testing | Is index testing/partner notification, with robust protections for patient confidentiality, provided for in national policy? |
| TP3—Compulsory testing | Is compulsory HIV testing prohibited under national law? |
| TP4—Age restrictions on testing & treatment | Can adolescents access HIV testing and treatment without parental consent under national policy? |
| TP5—PrEP | Are all people/populations at substantial risk of HIV infection eligible for PrEP under national policy and have technologies for pre-exposure prophylaxis (PrEP) received national regulatory approval? |
| TP6—Harm reduction | Does national law and HIV policy incorporate key harm reduction strategies, including avoidance of criminalizing syringe possession? |
| TP7—Comprehensive sexuality education | Is comprehensive sexuality education required in primary and secondary schools under national policy? |
| TP8—Prisoners prevention | Are both condoms/lubricants and syringe access/exchange programs available to prisoners as a matter of policy? |

Source: HIV Policy Lab (https://www.hivpolicylab.org)

## Approach

**Temporal dimensions.** The OxCGRT has been tracking the COVID-19 response since March of 2020 when the majority of countries began rapidly introducing COVID-19 countermeasures and has been reporting changes in the major countermeasures on a daily basis since then. Based on news reports, the data go back to January of 2020. It therefore represents a country-day panel. In order to examine associations between the COVID-19 policy responses and the HIV policy and examine how the response has evolved over time, we divide the COVID-19 Policy Measures Index Scores into three time periods from the time when the first case was discovered in a country: $< = 30$ days; $< = 365$ days; and $< = 730$ days. We then also examine the associations with monthly averages of the COVID-19 Policy Measures Index Scores. This approach recognizes that the response may have evolved since the beginning of the pandemic. While at the beginning of the pandemic all countries converged relatively rapidly towards a similar set of lock-down measures [45] as time went on, countries have diverged more in their policy responses in ways that may align with policy learning from HIV. For the HIV Policy Index Scores, we construct an average of the Index over the period 2017–2020 and treat the overall totality of the policy response to HIV as relatively static.

We first summarize descriptively which countries had the most and least aggressive COVID-19 policy responses and HIV policy responses based on total scores and compare them. Next, using OLS regression models, we estimated the effect of HIV Policy Scores on the

**Table 2.** Summary COVID-19 policy response measures.

| Index Name | Policies Included |
|---|---|
| Containment and Closure Measures | School closing, Workplace closing, Cancel public events, Restrictions on gathering size, Close public transport, Stay-at-home requirements, Restrictions on internal movement, Restrictions on international travel |
| Economic Support Measures | Income support, Debt/contract relief for households |
| Health Systems Measures | Public information campaign, Testing policy, Contact tracing, Facial coverings, Vaccination policy |

Source: Reproduced from Hale et al. (2021)

COVID-19 Policy Scores for three time points. In one set of models, we included all 158 countries; in the other set of models, we restricted the countries to low- and middle- income countries (LMICs). We stratify by HICs and LMICs as HICs by nature of their higher-income status are likely to have access to more resources and capacity to address both HIV and COVID-19. The country income groups came from the World Bank as of 2019 [51]. We controlled for the universal health coverage index, HIV prevalence, and GDP per capita (logged), and the polity regime scores for 2019 for both sets of models. Finally, we regressed the HIV Policy Index Scores against the monthly averages of the COVID-19 Policy Scores with controlling for the three covariates. We present the results using monthly coefficient plots to see how these relationships have evolved over time.

## Results

### Summary statistics of COVID-19 policy response and HIV policy response

Table 3 presents the descriptive statistics of HIV and COVID-19 policy response scores. As time went on, the mean value of the Containment and Closure score went down, whereas Economic Support or Health Systems scores did not show significant changes. Among LMICs, these patterns were similar (see S3 Table). S1 Table summarizes the top and bottom ten HIV and COVID-19 policy response scores by countries. A few general trends are observable. First, countries that are strong in their HIV policies are not necessarily the countries that were the strongest in the COVID-19 policies with the exception of Rwanda, which scored in the top 10

**Table 3. Descriptive statistics.**

| Variable | Obs. | Mean | Std. dev. | Min | Max |
|---|---|---|---|---|---|
| **HIV Policy Index Scores** | | | | | |
| Overall | 146 | 58.03 | 12.33 | 15 | 86 |
| Testing and Prevention | 146 | 49.00 | 21.32 | 0 | 100 |
| Health Systems | 143 | 63.24 | 20.36 | 0 | 100 |
| Structural | 146 | 51.83 | 16.26 | 11.1 | 88.9 |
| Clinical Treatment | 145 | 71.79 | 17.97 | 28.6 | 100 |
| **COVID-19 Policy Index Scores: First 30 days from the first care reporting** | | | | | |
| Containment and Closure | 145 | 42.60 | 24.04 | 0 | 90.08 |
| Economic Support | 145 | 14.82 | 17.47 | 0 | 91.67 |
| Health Systems | 146 | 35.63 | 12.96 | 0 | 59.78 |
| **COVID-19 Policy Index Scores: First 365 days (1 year) from the first case reporting** | | | | | |
| Containment and Closure | 145 | 53.55 | 14.17 | 3.89 | 86.20 |
| Economic Support | 145 | 45.77 | 23.48 | 0 | 99.32 |
| Health Systems | 146 | 57.23 | 9.70 | 19.99 | 74.96 |
| **COVID-19 Policy Index Scores: First 730 days (2 years) from the first case reporting** | | | | | |
| Containment and Closure | 145 | 50.06 | 13.21 | 3.63 | 73.40 |
| Economic Support | 145 | 41.88 | 23.85 | 0 | 99.66 |
| Health Systems | 146 | 67.37 | 10.09 | 19.99 | 83.44 |
| **Covariates** | | | | | |
| UHC service coverage index | 145 | 64.42 | 16.12 | 27.33 | 89.36 |
| HIV Prevalence | 145 | 1.60 | 4.21 | 0.002 | 27.34 |
| GDP per capita (logged) | 145 | 8.51 | 1.45 | 5.61 | 11.66 |
| Polity Score (-10: strongly autocratic, +10: strongly democratic) | 145 | 4.87 | 5.61 | -10 | 10 |

**Table 4. All countries, containment and closure for COVID-19, three time points.**

| HIV Policy Scores (2019) | Containment and Closure | | | | | | | | | | | | | | |
| --- | --- | --- | --- | --- | --- | --- | --- | --- | --- | --- | --- | --- | --- | --- | --- |
| | First 30 days | | | | | First 1 year (365 days) cumulative | | | | | First 2 years (730 days) cumulative | | | | |
| | (1) | (2) | (3) | (4) | (5) | (6) | (7) | (8) | (9) | (10) | (11) | (12) | (13) | (14) | (15) |
| Testing and Prevention | -0.269** | | | | -0.174 | -0.193*** | | | | -0.214*** | -0.155** | | | | -0.131* |
| | (0.095) | | | | (0.106) | (0.057) | | | | (0.063) | (0.053) | | | | (0.058) |
| Health Systems | | -0.207 | | | -0.161 | | -0.114 | | | -0.084 | | -0.183** | | | -0.160* |
| | | (0.106) | | | (0.119) | | (0.064) | | | (0.071) | | (0.058) | | | (0.066) |
| Structural | | | 0.118 | | 0.235 | | | 0.059 | | 0.171 | | | -0.014 | | 0.100 |
| | | | (0.149) | | (0.159) | | | (0.091) | | (0.095) | | | (0.083) | | (0.087) |
| Clinical Treatment | | | | -0.302** | -0.220 | | | | 0.027 | 0.098 | | | | 0.005 | 0.072 |
| | | | | (0.112) | (0.120) | | | | (0.070) | (0.072) | | | | (0.064) | (0.066) |
| UHC service coverage index | -0.300 | -0.399 | -0.470 | -0.486* | -0.544* | 0.387** | 0.295* | 0.280 | 0.335* | 0.312* | 0.362** | 0.281* | 0.310* | 0.321* | 0.302* |
| | (0.230) | (0.233) | (0.248) | (0.232) | (0.257) | (0.140) | (0.142) | (0.152) | (0.145) | (0.153) | (0.129) | (0.127) | (0.139) | (0.132) | (0.141) |
| HIV Prevalence | 0.851 | 0.724 | 0.491 | 0.848 | 1.247* | 0.602* | 0.472 | 0.337 | 0.303 | 0.664* | 0.383 | 0.412 | 0.155 | 0.170 | 0.511 |
| | (0.468) | (0.479) | (0.461) | (0.471) | (0.497) | (0.284) | (0.291) | (0.283) | (0.295) | (0.296) | (0.261) | (0.261) | (0.258) | (0.268) | (0.273) |
| GDP per capita (log) | -1.610 | -0.940 | -1.364 | -1.494 | -0.227 | -0.712 | -0.229 | -0.602 | -0.882 | 0.019 | 0.013 | 0.894 | -0.055 | -0.162 | 0.975 |
| | (2.596) | (2.706) | (2.692) | (2.616) | (2.722) | (1.574) | (1.645) | (1.651) | (1.637) | (1.622) | (1.449) | (1.474) | (1.508) | (1.486) | (1.497) |
| Polity Score | 0.207 | 0.249 | -0.104 | 0.048 | 0.041 | 0.002 | 0.004 | -0.190 | -0.124 | -0.099 | -0.118 | -0.044 | -0.197 | -0.221 | -0.108 |
| | (0.355) | (0.375) | (0.403) | (0.352) | (0.404) | (0.216) | (0.228) | (0.247) | (0.221) | (0.241) | (0.198) | (0.204) | (0.226) | (0.200) | (0.222) |
| Constant | 86.412*** | 87.153*** | 78.075*** | 106.677*** | 99.817*** | 43.135*** | 42.877*** | 37.981*** | 37.521*** | 32.499*** | 34.201*** | 35.324*** | 32.002*** | 30.992*** | 28.038** |
| | (12.364) | (12.911) | (13.437) | (15.395) | (15.937) | (7.499) | (7.849) | (8.244) | (9.636) | (9.499) | (6.900) | (7.032) | (7.528) | (8.747) | (8.763) |
| Observations | 145 | 142 | 145 | 144 | 141 | 145 | 142 | 145 | 144 | 141 | 145 | 142 | 145 | 144 | 141 |
| R-squared | 0.195 | 0.170 | 0.152 | 0.191 | 0.228 | 0.147 | 0.096 | 0.081 | 0.083 | 0.188 | 0.170 | 0.179 | 0.118 | 0.125 | 0.213 |

Standard errors in parentheses

*** $p<0.001$, ** $p<0.01$, * $p<0.05$

for each. There was also little similarity in the bottom 10 countries with the exception of Yemen. A more detailed list of country rankings is in the S2 Table.

**All countries.** In Table 4, we generally find negative associations between the HIV Testing and Prevention measures and Containment and Closure measures throughout the study time period. Although it is not significant when the model is controlled for the other HIV policy scores for the first 30 days (see column 4), it is consistently significant for the first 365 days and 730 days. In Table 5, we find consistent positive associations of HIV Health Systems scores and HIV Structural scores with COVID-19 Economic Support scores. Especially, the Structural score were associated with Economic Support measures for 365 days and beyond (see column 8, 10, 13, and 15). Clinical Treatment shows significant negative association with Economic Support measures in the first 30 days but is not for longer time span. Finally, in Table 6, we find that Testing and Prevention policy score for HIV is negatively associated with Health Systems measures for the first 365 days and beyond (see column 6,10, 11, and 15).

**Low-income countries.** Tables 7–9 present the regression results from LMICs. For LMICs, the associations between HIV policy scores for HIV and Economic Support measures for COVID-19 show similar patterns with all-countries models (Table 8). Clinical Treatment

**Table 5. All countries, economic support for COVID-19, three time points.**

| HIV Policy Scores (2019) | First 30 days | | | | | First 1 year (365 days) cumulative | | | | | First 2 years (730 days) cumulative | | | | |
|---|---|---|---|---|---|---|---|---|---|---|---|---|---|---|---|
| | (1) | (2) | (3) | (4) | (5) | (6) | (7) | (8) | (9) | (10) | (11) | (12) | (13) | (14) | (15) |
| Testing and Prevention | 0.035 | | | | | -0.056 | | | | | -0.044 | | | | -0.034 |
| | (0.075) | | | | | (0.079) | | | | | (0.080) | | | | (0.088) |
| Health Systems | | 0.083 | | | | | 0.099 | | | | | 0.084 | | | 0.052 |
| | | (0.082) | | | | | (0.086) | | | | | (0.087) | | | (0.099) |
| Structural | | | 0.193 | | 0.182 | | | 0.399*** | | 0.437*** | | | 0.310* | | 0.336* |
| | | | (0.113) | | (0.121) | | | (0.116) | | (0.127) | | | (0.119) | | (0.131) |
| Clinical Treatment | | | | -0.275** | -0.339*** | | | | -0.183* | -0.221* | | | | -0.108 | -0.133 |
| | | | | (0.085) | (0.092) | | | | (0.092) | (0.096) | | | | (0.094) | (0.100) |
| UHC service coverage index | -0.093 | -0.061 | -0.188 | -0.175 | -0.319 | 0.379 | 0.360 | 0.132 | 0.298 | 0.047 | 0.503* | 0.484* | 0.311 | 0.451* | 0.259 |
| | (0.182) | (0.181) | (0.189) | (0.176) | (0.195) | (0.192) | (0.190) | (0.194) | (0.191) | (0.204) | (0.194) | (0.192) | (0.199) | (0.194) | (0.212) |
| HIV Prevalence | 0.682 | 0.613 | 0.780* | 1.064** | 0.968* | -0.411 | -0.600 | -0.394 | -0.267 | -0.112 | -0.644 | -0.788* | -0.632 | -0.576 | -0.443 |
| | (0.370) | (0.371) | (0.351) | (0.358) | (0.377) | (0.390) | (0.391) | (0.360) | (0.388) | (0.395) | (0.394) | (0.393) | (0.370) | (0.395) | (0.410) |
| GDP per capita (log) | 0.919 | 0.431 | 1.419 | 1.241 | 1.339 | 5.628* | 5.352* | 6.637** | 5.787** | 6.853** | 4.218 | 4.079 | 5.002* | 4.312 | 5.206* |
| | (2.050) | (2.095) | (2.052) | (1.984) | (2.068) | (2.165) | (2.208) | (2.104) | (2.151) | (2.167) | (2.187) | (2.224) | (2.160) | (2.191) | (2.250) |
| Polity Score | 0.385 | 0.331 | 0.172 | 0.421 | 0.083 | 0.465 | 0.303 | -0.056 | 0.438 | -0.159 | 0.466 | 0.320 | 0.062 | 0.444 | -0.028 |
| | (0.281) | (0.291) | (0.307) | (0.267) | (0.307) | (0.296) | (0.306) | (0.315) | (0.290) | (0.322) | (0.299) | (0.308) | (0.323) | (0.295) | (0.334) |
| Constant | 8.292 | 7.373 | 2.782 | 31.68** | 28.321* | -25.349* | -29.941** | -38.953*** | -11.230 | -24.023 | -25.477* | -29.881** | -36.049** | -17.340 | -28.124* |
| | (9.765) | (9.996) | (10.24) | (11.676) | (12.109) | (10.313) | (10.536) | (10.504) | (12.659) | (12.691) | (10.416) | (10.610) | (10.783) | (12.898) | (13.173) |
| Observations | 145 | 142 | 145 | 144 | 141 | 145 | 142 | 145 | 144 | 141 | 145 | 142 | 145 | 144 | 141 |
| R-squared | 0.050 | 0.055 | 0.068 | 0.116 | 0.151 | 0.413 | 0.421 | 0.457 | 0.427 | 0.488 | 0.420 | 0.429 | 0.445 | 0.424 | 0.464 |

Standard errors in parentheses
*** $p < 0.001$, ** $p < 0.01$, * $p < 0.05$

**Table 6. All countries, health systems for COVID-19, three time points.**

| HIV Policy Scores (2019) | First 30 days | | | | | First 1 year (365 days) cumulative | | | | | First 2 years (730 days) cumulative | | | | |
|---|---|---|---|---|---|---|---|---|---|---|---|---|---|---|---|
| | (1) | (2) | (3) | (4) | (5) | (6) | (7) | (8) | (9) | (10) | (11) | (12) | (13) | (14) | (15) |
| Testing and Prevention | -0.053 | | | | -0.005 | -0.100* | | | | -0.097* | -0.101* | | | | -0.099* |
| | (0.056) | | | | (0.062) | (0.040) | | | | (0.045) | (0.039) | | | | (0.045) |
| Health Systems | | -0.117 | | | -0.138 | | -0.065 | | | -0.057 | | -0.054 | | | -0.038 |
| | | (0.060) | | | (0.070) | | (0.044) | | | (0.051) | | (0.044) | | | (0.050) |
| Structural | | | 0.032 | | 0.087 | | | 0.029 | | 0.059 | | | 0.009 | | 0.041 |
| | | | (0.085) | | (0.093) | | | (0.062) | | (0.067) | | | (0.061) | | (0.067) |
| Clinical Treatment | | | | -0.063 | -0.037 | | | | -0.017 | 0.029 | | | | -0.033 | 0.010 |
| | | | | (0.066) | (0.071) | | | | (0.048) | (0.051) | | | | (0.047) | (0.051) |
| UHC service coverage index | -0.135 | -0.167 | -0.173 | -0.174 | -0.233 | -0.033 | -0.075 | -0.085 | -0.075 | -0.070 | 0.019 | -0.024 | -0.024 | -0.029 | -0.015 |
| | (0.135) | (0.133) | (0.142) | (0.136) | (0.150) | (0.096) | (0.097) | (0.103) | (0.099) | (0.108) | (0.095) | (0.096) | (0.102) | (0.098) | (0.107) |
| HIV Prevalence | 0.388 | 0.491 | 0.320 | 0.391 | 0.590* | 0.309 | 0.262 | 0.173 | 0.187 | 0.363 | 0.417* | 0.356 | 0.275 | 0.314 | 0.468* |
| | (0.276) | (0.273) | (0.266) | (0.278) | (0.293) | (0.196) | (0.200) | (0.193) | (0.202) | (0.211) | (0.194) | (0.198) | (0.191) | (0.200) | (0.209) |
| GDP per capita (log) | 1.335 | 2.020 | 1.395 | 1.365 | 2.409 | 2.533* | 2.861* | 2.571* | 2.514* | 3.027** | 2.857** | 3.139** | 2.847* | 2.853* | 3.244** |
| | (1.532) | (1.544) | (1.550) | (1.540) | (1.599) | (1.090) | (1.131) | (1.124) | (1.120) | (1.154) | (1.079) | (1.120) | (1.114) | (1.108) | (1.142) |
| Polity Score | 0.179 | 0.229 | 0.109 | 0.150 | 0.155 | 0.135 | 0.123 | 0.041 | 0.076 | 0.106 | 0.229 | 0.210 | 0.158 | 0.171 | 0.207 |
| | (0.209) | (0.213) | (0.230) | (0.207) | (0.236) | (0.148) | (0.156) | (0.167) | (0.150) | (0.170) | (0.147) | (0.155) | (0.166) | (0.149) | (0.169) |
| Constant | 34.13*** | 34.45*** | 32.18*** | 38.38*** | 35.41*** | 41.58*** | 40.67*** | 38.91*** | 41.25*** | 38.05*** | 45.06*** | 44.00*** | 43.00*** | 46.01*** | 43.30*** |
| | (7.305) | (7.373) | (7.759) | (9.069) | (9.393) | (5.197) | (5.404) | (5.625) | (6.595) | (6.779) | (5.145) | (5.348) | (5.576) | (6.525) | (6.709) |
| Observations | 146 | 143 | 146 | 145 | 142 | 146 | 143 | 146 | 145 | 142 | 146 | 143 | 146 | 145 | 142 |
| R-squared | 0.032 | 0.052 | 0.026 | 0.032 | 0.063 | 0.125 | 0.103 | 0.087 | 0.086 | 0.140 | 0.209 | 0.186 | 0.172 | 0.175 | 0.219 |

Standard errors in parentheses

*** p<0.001, ** p<0.01, * p<0.05

**Table 7. Low- and middle-income countries, containment and closure for COVID-19, three time points.**

| HIV Policy Scores | Containment and Closure | | | | | | | | | | | | | | |
|---|---|---|---|---|---|---|---|---|---|---|---|---|---|---|---|
| | First 30 days | | | | | First 1 year (365 days) cumulative | | | | | First 2 years (730 days) cumulative | | | | |
| | (1) | (2) | (3) | (4) | (5) | (6) | (7) | (8) | (9) | (10) | (11) | (12) | (13) | (14) | (15) |
| Testing and Prevention | -0.190 | | | | -0.138 | -0.106 | | | | -0.156 | -0.067 | | | | -0.056 |
| | (0.131) | | | | (0.147) | (0.079) | | | | (0.086) | (0.073) | | | | (0.079) |
| Health Systems | | -0.143 | | | -0.154 | | -0.074 | | | -0.093 | | -0.179* | | | -0.182* |
| | | (0.143) | | | (0.167) | | (0.086) | | | (0.097) | | (0.077) | | | (0.090) |
| Structural | | | 0.060 | | 0.130 | | | 0.055 | | 0.171 | | | -0.048 | | 0.068 |
| | | | (0.186) | | (0.218) | | | (0.112) | | (0.126) | | | (0.102) | | (0.117) |
| Clinical Treatment | | | | -0.176 | -0.112 | | | | 0.137 | 0.160 | | | | 0.065 | 0.082 |
| | | | | (0.155) | (0.170) | | | | (0.093) | (0.099) | | | | (0.085) | (0.092) |
| UHC service coverage index | -0.356 | -0.447 | -0.488 | -0.500 | -0.477 | 0.198 | 0.135 | 0.114 | 0.188 | 0.183 | 0.158 | 0.112 | 0.145 | 0.157 | 0.138 |
| | (0.271) | (0.265) | (0.278) | (0.268) | (0.307) | (0.163) | (0.159) | (0.167) | (0.160) | (0.178) | (0.150) | (0.143) | (0.153) | (0.147) | (0.165) |
| HIV Prevalence | 0.719 | 0.622 | 0.450 | 0.667 | 1.008 | 0.406 | 0.349 | 0.259 | 0.084 | 0.440 | 0.191 | 0.354 | 0.081 | 0.024 | 0.355 |
| | (0.517) | (0.526) | (0.485) | (0.522) | (0.606) | (0.311) | (0.315) | (0.291) | (0.312) | (0.352) | (0.286) | (0.283) | (0.266) | (0.286) | (0.326) |
| GDP per capita (log) | 2.973 | 4.518 | 4.472 | 3.727 | 3.127 | 5.147* | 5.798* | 5.992* | 6.269* | 5.064* | 5.938* | 6.447** | 6.454** | 6.395** | 6.205** |
| | (4.116) | (4.051) | (4.026) | (4.086) | (4.281) | (2.478) | (2.420) | (2.419) | (2.439) | (2.486) | (2.278) | (2.180) | (2.212) | (2.239) | (2.306) |
| Polity Score | 0.265 | 0.291 | 0.163 | 0.182 | 0.186 | 0.096 | 0.111 | 0.019 | 0.093 | 0.030 | -0.093 | -0.061 | -0.063 | -0.105 | -0.094 |
| | (0.436) | (0.448) | (0.473) | (0.440) | (0.495) | (0.262) | (0.267) | (0.284) | (0.263) | (0.287) | (0.241) | (0.241) | (0.260) | (0.241) | (0.266) |
| Constant | 51.551* | 44.955 | 37.053 | 58.690* | 65.798* | 6.059 | 4.245 | -2.472 | -16.203 | -4.495 | -2.555 | 2.912 | -6.334 | -13.536 | -3.174 |
| | (23.450) | (23.345) | (22.067) | (28.098) | (30.252) | (14.118) | (13.948) | (13.257) | (16.775) | (17.571) | (12.975) | (12.565) | (12.126) | (15.397) | (16.300) |
| Observations | 103 | 100 | 103 | 102 | 99 | 103 | 100 | 103 | 102 | 99 | 103 | 100 | 103 | 102 | 99 |
| R-squared | 0.067 | 0.055 | 0.048 | 0.059 | 0.074 | 0.227 | 0.213 | 0.214 | 0.229 | 0.266 | 0.257 | 0.287 | 0.252 | 0.255 | 0.289 |

Standard errors in parentheses

*** p<0.001, ** p<0.01, * p<0.05

score is negatively associated in the first 30 days but is not for longer time span (Column 5 in Table 8), whereas Structural policy score has positive association for the first 365+ days (see columns 8, 10, 13, and 15 in Table 8). Lastly, in Table 9, Health Systems measure for COVID-19 is negatively associated with Health Systems scores for HIV but positively associated with Structural policy scores for HIV only for the first 30 days.

**Monthly COVID-19 policy scores for all countries.** These regression results allow us to observe in a more granular manner the change in relationship between the HIV Policy Indices and the COVID-19 Policy Measures Indices. As presented in Fig 1, we find that monthly Containment and Closure Measures Index were mostly negatively associated with HIV Testing and Prevention Scores and Health Systems Scores in early 2020, but by early 2021, the relationship reverses. Although the Structural dimension of HIV Policy Indices shows positive association with Containment and Closure Measures for COVID-19 in early 2020, the relationship shows similar pattern with those of HIV Testing and Prevention and Health Systems

**Table 8. Low- and middle-income countries, economic support for COVID-19, three time points.**

| HIV Policy Scores | Economic Support | | | | | | | | | | | | | | |
|---|---|---|---|---|---|---|---|---|---|---|---|---|---|---|---|
| | First 30 days | | | | | First 1 year (365 days) cumulative | | | | | First 2 years (730 days) cumulative | | | | |
| | (1) | (2) | (3) | (4) | (5) | (6) | (7) | (8) | (9) | (10) | (11) | (12) | (13) | (14) | (15) |
| Testing and Prevention | 0.049 | | | | 0.099 | -0.100 | | | | -0.097 | -0.081 | | | | -0.080 |
| | (0.091) | | | | (0.100) | (0.106) | | | | (0.114) | (0.103) | | | | (0.111) |
| Health Systems | | 0.040 | | | -0.025 | | 0.042 | | | -0.060 | | -0.027 | | | -0.124 |
| | | (0.099) | | | (0.113) | | (0.115) | | | (0.130) | | (0.110) | | | (0.126) |
| Structural | | | 0.157 | | 0.155 | | | 0.360* | | 0.458** | | | 0.283* | | 0.393* |
| | | | (0.127) | | (0.147) | | | (0.145) | | (0.169) | | | (0.141) | | (0.164) |
| Clinical Treatment | | | | -0.207 | -0.242* | | | | -0.109 | -0.128 | | | | -0.024 | -0.034 |
| | | | | (0.105) | (0.115) | | | | (0.125) | (0.132) | | | | (0.121) | (0.128) |
| UHC service coverage index | -0.030 | 0.007 | -0.078 | -0.073 | -0.200 | 0.407 | 0.351 | 0.182 | 0.319 | 0.140 | 0.581** | 0.531* | 0.403 | 0.530* | 0.368 |
| | (0.188) | (0.183) | (0.190) | (0.182) | (0.208) | (0.219) | (0.214) | (0.217) | (0.216) | (0.239) | (0.211) | (0.204) | (0.211) | (0.209) | (0.231) |
| HIV Prevalence | 0.709 | 0.715 | 0.816* | 1.034** | 0.999* | -0.303 | -0.486 | -0.374 | -0.316 | 0.064 | -0.514 | -0.562 | -0.573 | -0.604 | -0.182 |
| | (0.359) | (0.364) | (0.332) | (0.354) | (0.410) | (0.418) | (0.424) | (0.378) | (0.421) | (0.470) | (0.403) | (0.405) | (0.368) | (0.407) | (0.456) |
| GDP per capita (log) | 0.230 | -0.082 | -0.122 | -0.648 | -0.041 | 5.217 | 6.105 | 6.075 | 5.685 | 4.846 | 2.539 | 3.285 | 3.229 | 3.104 | 2.538 |
| | (2.862) | (2.799) | (2.753) | (2.770) | (2.896) | (3.331) | (3.261) | (3.141) | (3.295) | (3.326) | (3.212) | (3.119) | (3.057) | (3.186) | (3.221) |
| Polity Score | 0.134 | 0.143 | -0.003 | 0.111 | -0.052 | 0.435 | 0.354 | 0.068 | 0.391 | -0.041 | 0.435 | 0.370 | 0.147 | 0.412 | 0.052 |
| | (0.303) | (0.309) | (0.324) | (0.298) | (0.335) | (0.353) | (0.360) | (0.369) | (0.355) | (0.384) | (0.340) | (0.345) | (0.359) | (0.343) | (0.372) |
| Constant | 9.739 | 10.023 | 10.152 | 35.820 | 31.295 | -22.360 | -32.560 | -36.366* | -17.235 | -12.824 | -16.270 | -21.252 | -27.412 | -19.225 | -12.902 |
| | (16.302) | (16.129) | (15.091) | (19.049) | (20.466) | (18.975) | (18.789) | (17.216) | (22.660) | (23.504) | (18.297) | (17.971) | (16.754) | (21.906) | (22.763) |
| Observations | 103 | 100 | 103 | 102 | 99 | 103 | 100 | 103 | 102 | 99 | 103 | 100 | 103 | 102 | 99 |
| R-squared | 0.061 | 0.058 | 0.072 | 0.093 | 0.109 | 0.261 | 0.255 | 0.299 | 0.260 | 0.320 | 0.288 | 0.283 | 0.312 | 0.283 | 0.329 |

Standard errors in parentheses

*** p<0.001, ** p<0.01, * p<0.05

Scores. Interestingly, we see a reverse pattern of coefficient between Containment and Closure Measures for COVID-19 and HIV Clinical Treatment Scores.

In Fig 2, the Economic Support Measures Index shows a somewhat different pattern. Throughout the pandemic economic support measures were positively associated with Health Systems and Structural policy measures. However, Testing and Prevention policy measure were positively associated with Economic Support measures in early 2020, and it shows marginal changes around zero over time. A countries' HIV Clinical Treatment Score was mostly negatively associated with economic support measures most of the time in 2020 but in 2021, it the association became similar to Testing and Prevention measures.

As seen in Fig 3, monthly Health Systems Measures Index for COVID-19 and HIV Testing and Prevention and Health Systems policy scores are mostly negatively associated throughout the pandemic. In contrast, we see that the coefficients for the association between HIV Clinical Treatment scores and Health Systems Measures for COVID-19 become more negative as time

**Table 9. Low- and middle-income countries, health systems for COVID-19, three time points.**

| HIV Policy Scores | Health Systems | | | | | | | | | | | | | | |
| | First 30 days | | | | | First 1 year (365 days) cumulative | | | | | First 2 years (730 days) cumulative | | | | |
| | (1) | (2) | (3) | (4) | (5) | (6) | (7) | (8) | (9) | (10) | (11) | (12) | (13) | (14) | (15) |
|---|---|---|---|---|---|---|---|---|---|---|---|---|---|---|---|
| Testing and Prevention | -0.003 | | | | 0.038 | -0.032 | | | | -0.051 | -0.050 | | | | -0.069 |
| | (0.073) | | | | (0.078) | (0.054) | | | | (0.060) | (0.054) | | | | (0.060) |
| Health Systems | | -0.132 | | | -0.212* | | -0.007 | | | -0.025 | | 0.002 | | | -0.002 |
| | | (0.078) | | | (0.089) | | (0.059) | | | (0.068) | | (0.059) | | | (0.068) |
| Structural | | | 0.121 | | 0.221 | | | 0.049 | | 0.039 | | | 0.013 | | 0.010 |
| | | | (0.101) | | (0.115) | | | (0.075) | | (0.088) | | | (0.076) | | (0.088) |
| Clinical Treatment | | | | -0.062 | -0.077 | | | | 0.085 | 0.112 | | | | 0.051 | 0.079 |
| | | | | (0.086) | (0.090) | | | | (0.063) | (0.069) | | | | (0.064) | (0.069) |
| UHC service coverage index | -0.161 | -0.170 | -0.218 | -0.179 | -0.323* | -0.091 | -0.108 | -0.131 | -0.090 | -0.075 | -0.017 | -0.044 | -0.049 | -0.034 | 0.003 |
| | (0.150) | (0.143) | (0.151) | (0.148) | (0.162) | (0.111) | (0.108) | (0.112) | (0.108) | (0.123) | (0.111) | (0.108) | (0.113) | (0.110) | (0.124) |
| HIV Prevalence | 0.309 | 0.517 | 0.333 | 0.383 | 0.718* | 0.196 | 0.169 | 0.160 | 0.037 | 0.128 | 0.336 | 0.270 | 0.266 | 0.194 | 0.271 |
| | (0.288) | (0.285) | (0.266) | (0.290) | (0.324) | (0.212) | (0.215) | (0.197) | (0.212) | (0.246) | (0.213) | (0.216) | (0.199) | (0.215) | (0.248) |
| GDP per capita (log) | 1.520 | 1.695 | 1.554 | 1.335 | 1.784 | 4.566** | 4.872** | 4.816** | 5.177** | 4.971** | 4.292* | 4.692** | 4.678** | 4.922** | 4.539* |
| | (2.295) | (2.196) | (2.208) | (2.267) | (2.285) | (1.689) | (1.655) | (1.637) | (1.660) | (1.735) | (1.696) | (1.660) | (1.650) | (1.680) | (1.750) |
| Polity Score | 0.335 | 0.337 | 0.223 | 0.324 | 0.148 | 0.310 | 0.287 | 0.257 | 0.320 | 0.296 | 0.414* | 0.386* | 0.392* | 0.415* | 0.414* |
| | (0.242) | (0.241) | (0.257) | (0.242) | (0.261) | (0.178) | (0.182) | (0.190) | (0.178) | (0.198) | (0.179) | (0.182) | (0.192) | (0.180) | (0.200) |
| Constant | 31.600* | 37.566** | 29.001* | 38.323* | 43.911** | 26.102** | 23.685* | 22.885* | 14.011 | 14.556 | 33.439*** | 29.652** | 29.658** | 23.896* | 25.232* |
| | (13.104) | (12.667) | (12.120) | (15.577) | (16.150) | (9.646) | (9.549) | (8.982) | (11.406) | (12.262) | (9.682) | (9.577) | (9.057) | (11.546) | (12.370) |
| Observations | 104 | 101 | 104 | 103 | 100 | 104 | 101 | 104 | 103 | 100 | 104 | 101 | 104 | 103 | 100 |
| R-squared | 0.050 | 0.078 | 0.064 | 0.055 | 0.123 | 0.157 | 0.155 | 0.158 | 0.170 | 0.185 | 0.210 | 0.203 | 0.203 | 0.209 | 0.222 |

Standard errors in parentheses

*** p<0.001, ** p<0.01, * p<0.05

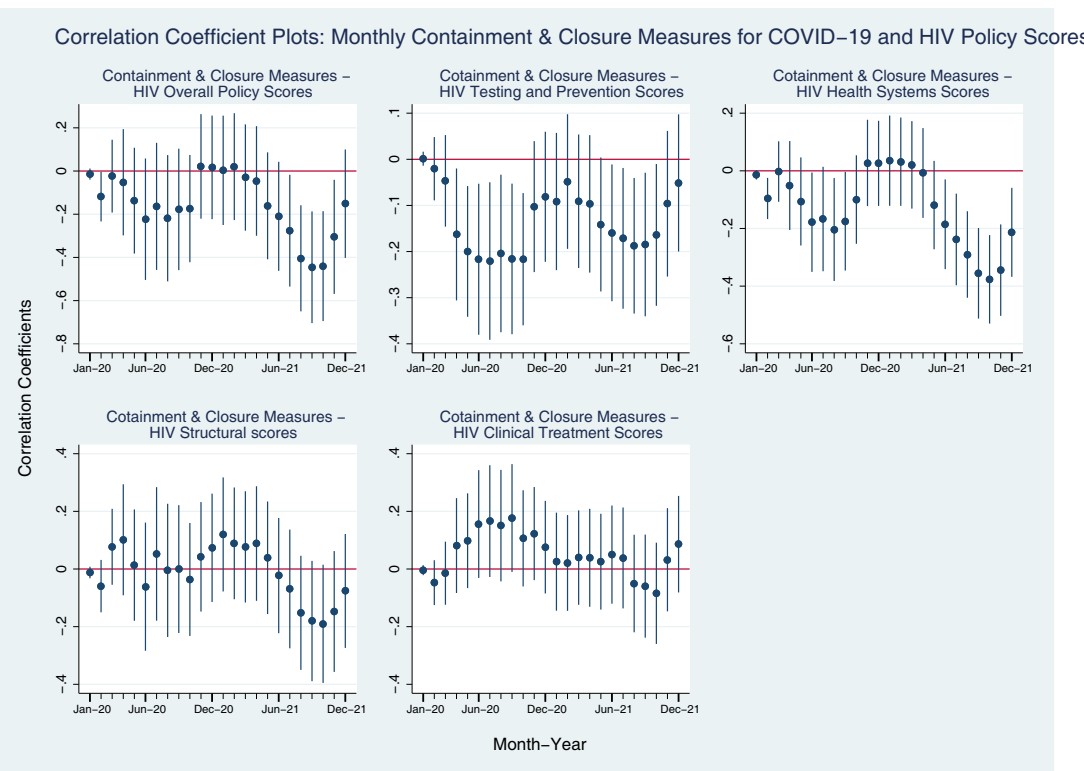

**Fig 1. Coefficient plots: Regression coefficients between COVID-19 containment and closure measures in each month from Jan 2020 to Dec 2021, all countries.** Note: Point estimates and 95% confidence intervals. N = 156 countries. All models are controlled for GDP per capita (logged), Universal Health Coverage Index, HIV Prevalence.

went on. Coefficients for HIV Structural policy scores are marginal changes around zero during the two-year of the pandemic.

**Monthly COVID-19 policy scores for LMICs.** We see very similar patterns between all-countries models and LMIC-models for the coefficients between the HIV policy scores and Economic Support Measures (Fig 4, compared to Fig 2). However, we find different patterns for the coefficients between HIV policy scores and Containment and Closure measures for COVID-19 (Fig 5, compared to Fig 1). In Fig 5, we see that while Testing and Prevention Scores for HIV among LMICs have positive associations with Containment and Closure for COVID-19 in early 2020, these associations become negative throughout the pandemic, and go then back to positive association at the end of 2021. In terms of Health Systems scores, LMICs consistently show negative associations with Containment and Closure measures for COVID-19, and the coefficients become more negative as time went on. Structural policy scores for HIV were positively associated with Containment and Closure measure for COVID-19 until the mid-2020, but the association turns negative throughout the rest of the pandemic. Interestingly, Clinical Treatment policy scores for HIV was positively associated with Containment and Closure measure for COVID-19 in LMICs, and the coefficients are even larger than the all-countries group model in 2020.

Lastly, in Fig 6, we mostly see positive associations between HIV Structural policy scores and Health Systems measures for COVID-19 throughout the pandemic in LMICs, whereas it was not always positive in all-countries model in Fig 3. Another big difference is found in the coefficient plots for Clinical Treatment policy scores for HIV and Health Systems measures for COVID-19 among LMICs. From the early-2020 to mid-2021, the association is mostly positive

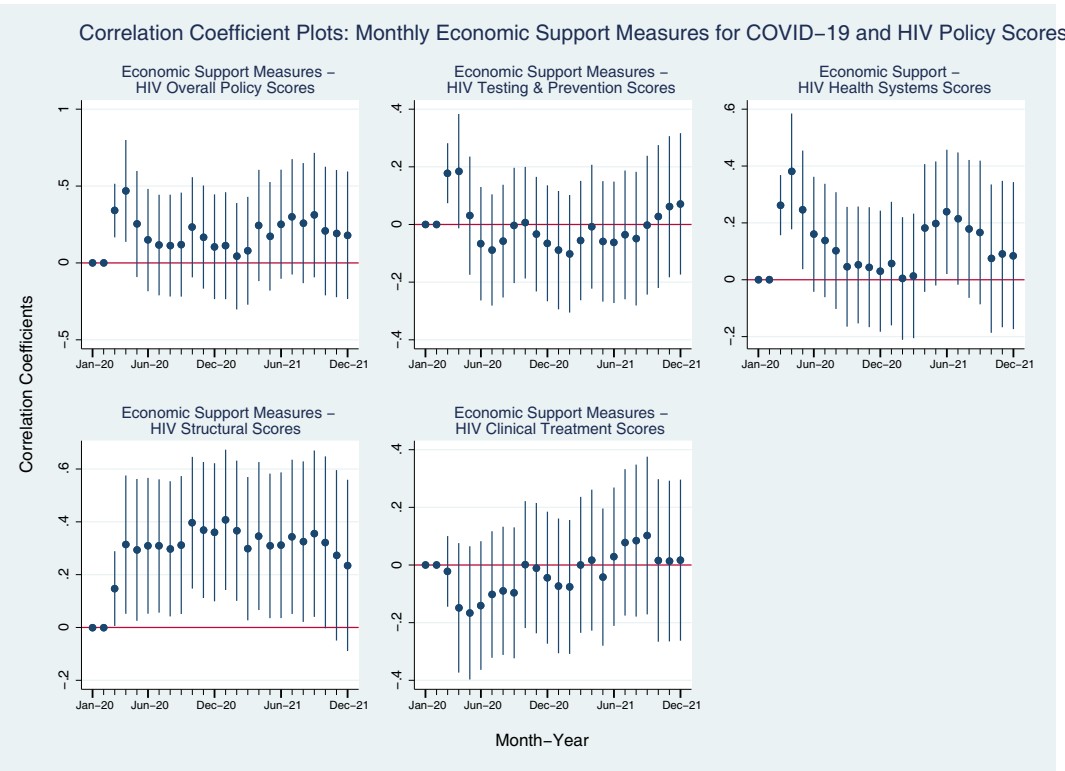

**Fig 2. Coefficient plots: Regression coefficients between COVID-19 economic support measures in each month from Jan 2020 to Dec 2021, all countries.** Note: Point estimates and 95% confidence intervals. N = 156 countries. All models are controlled for GDP per capita (logged), Universal Health Coverage Index, HIV Prevalence.

among LMICs, whereas it starts getting close to zero from the late 2020 for all-countries model.

## Discussion

We find evidence of associations between countries' prior HIV responses and their current COVID-19 responses but with some important nuance. Our findings partially support our hypothesis H2A by demonstrating higher and stronger association between COVID-19 Economic Support Measures and HIV Policy Indices over time. By contrast, Containment and Closure Measures were significantly *negatively* correlated with HIV Policy Indices in the first 30 days of the first case being detected suggesting an inverse relationship between use of stringent measures and HIV policy legacies. Health Systems Measures did not have significant correlations with HIV policy legacies, however. These results therefore also supported hypothesis H2B. Although we found that the HIV Policy Index Scores on "Structural Determinants" were positively associated with Economic Support Measures and were negatively associated with Containment and Closure Measures, we did not see their significant associations with Health Systems Measures.

That policy stringency (i.e., extent of lock-down measures) on its own was not associated with countries' HIV policy responses after the first 30 days, was telling. While all countries initially converged in this direction, more stringent policies, at least over a longer time frame, in many ways run counter to policy learning from HIV. As structural HIV policies include such policies as same-sex non-criminalization, sex work and drug use non-criminalization, national

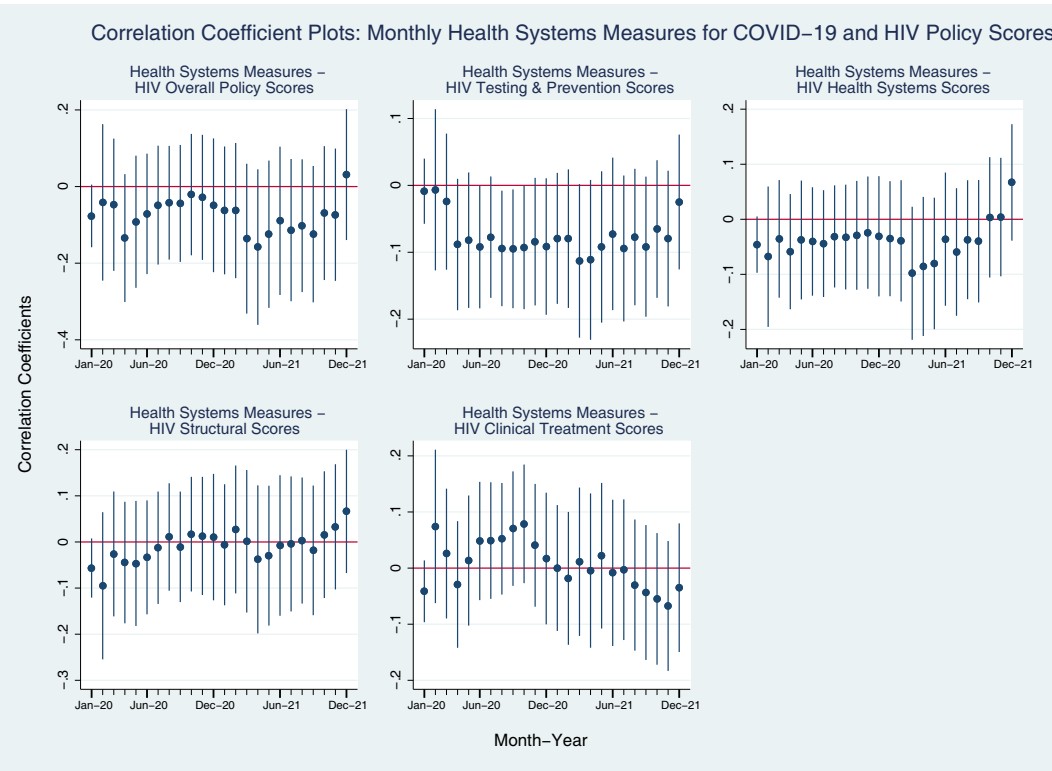

**Fig 3. Coefficient plots: Regression coefficients between COVID-19 health systems measures in each month from Jan 2020 to Dec 2021, all countries.** Note: Point estimates and 95% confidence intervals. N = 156 countries. All models are controlled for GDP per capita (logged), Universal Health Coverage Index, HIV Prevalence.

human rights institutions and non-discrimination protections, countries with stronger structural responses to HIV may be countries that respect human rights more and would be less inclined towards prolonged involuntary measures.

Up until COVID-19, public health had largely moved away from recommending highly restrictive policy responses, preferring disease responses that minimized incursions on individual rights considered effects on countries' economies [43]. For instance, Cuba's approach to testing and quarantining people with HIV in AIDS sanitoriums in the early years of the HIV pandemic were viewed as an overly reactionary and unnecessary infringement on individual rights [52]. AIDS activists and their public health allies initially sought approaches that would respect the autonomy and privacy rights of people with or at risk for HIV infection and that would offer protection from unwarranted discrimination. The belief that stigmatizing and compulsory measures would "drive the epidemic underground," was used to resist calls for compulsory measures such as isolation and quarantine, which had been so much a part of the public health armamentarium up until the time of HIV [53]. At the beginning of the pandemic, China's strict lock-down measures appeared to only be possible given its authoritarian disregard for individual rights and privacy. The WHO organization had a longstanding policy of encouraging countries to keep borders open to avoid 'unnecessary interference with international traffic,' which a number of countries quickly contravened [54]. While this may have been justified early in the pandemic to limit spread, many countries have maintained these restrictions throughout with more questionable justification. Of course, it is also possible that countries that moved too slowly to close borders and initiate lock-down measures also missed

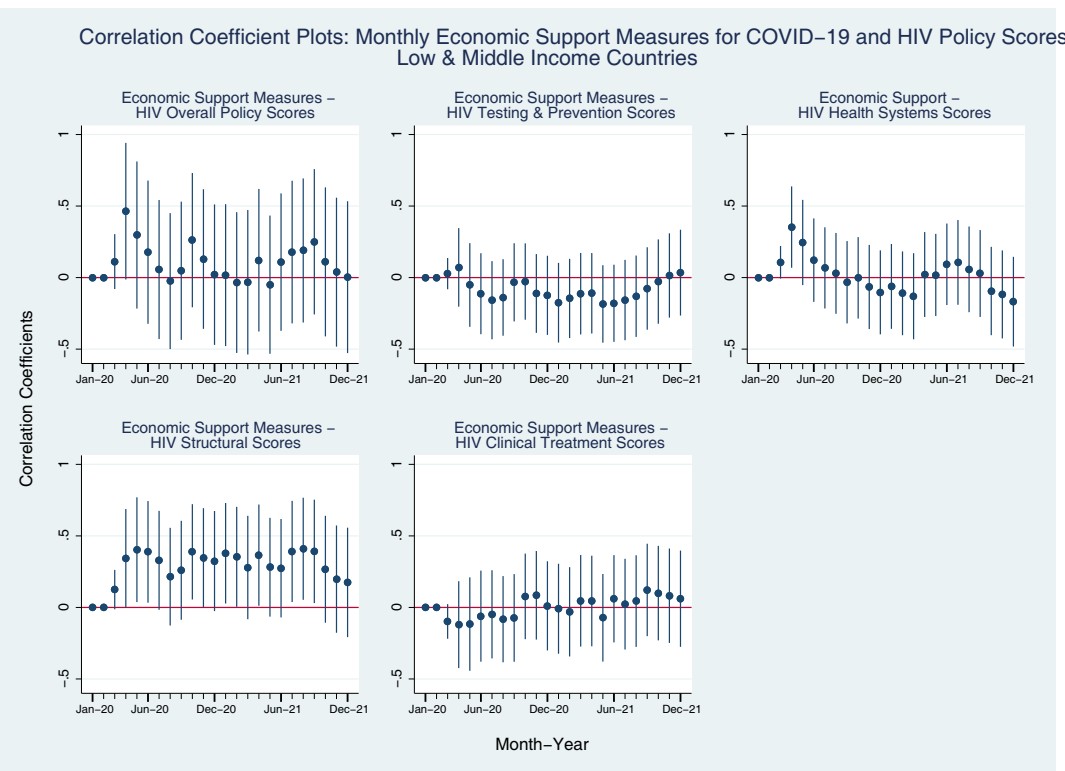

**Fig 4. Coefficient plots: Regression coefficients between COVID-19 economic support index (Y axis) and HIV policy scores (X axis) in each month from Jan 2020 to Dec 2021, low- and middle-income countries.** Note: Point estimates and 95% confidence intervals. N = 107 countries. All models are controlled for GDP per capita (logged), Universal Health Coverage Index, HIV Prevalence.

an opportunity to reduce early COVID-19 spread and catch existing cases in ways that may have contributed to the need for further lock-downs in order to "flatten the curve".

Sun (2020) suggests that once the initial COVID-19 pandemic subsides, we will need a period of reflection to determine whether the Siracusa principles need to be updated or potentially a General Comment developed to solidify human rights standards specifying how rights-limiting steps may be operationalized related to pandemic preparedness [28]. It could, for example, clarify key considerations such as when movement, free speech, and peaceful assembly may be rightfully limited and when they should not. Overall, what COVID-19 has demonstrated is the need to adapt strategies in real time as new information arises. Thus, one lesson is that future pandemic responses will need to be adaptable to changing conditions while keeping in mind principles that we have learned from the HIV response. These include the importance of developing evidence-based interventions that also account for community participation, attention to avoiding stigma and protecting marginalized and vulnerable and utilizing the least restrictive means available to achieve a public health end.

Meanwhile, countries that scored higher on their HIV Clinical and Treatment Scores, scored lower on their COVID-19 Economic Support Index. This suggests that countries with a strong biomedical infrastructure may be less prepared for some of the social elements that made this pandemic not only a public health crisis but also an economic crisis, with its own public health implications. HIV has contributed greatly to public health's collective understanding of the importance of the intersections of economic security and health security. For instance, women's economic insecurity has served as a systemic barrier to behavioral

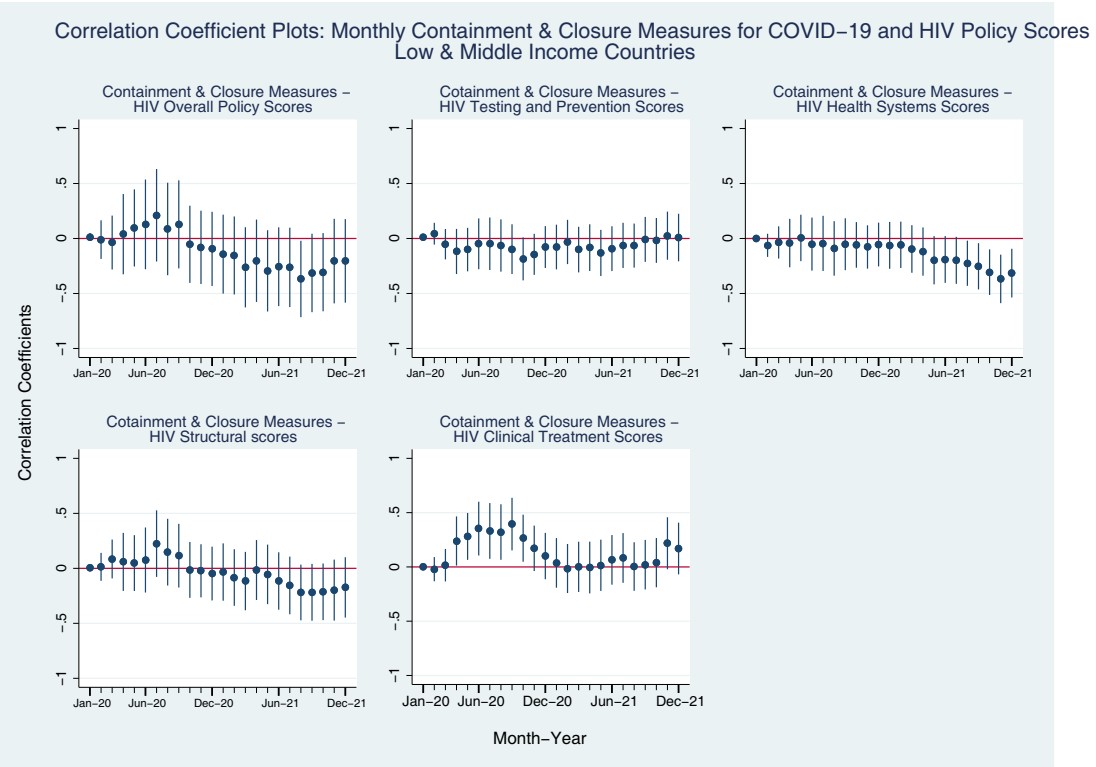

**Fig 5. Coefficient plots: Regression coefficients between COVID-19 containment and closure index (Y axis) and HIV policy scores (X axis) in each month from Jan 2020 to Dec 2021, low- and middle-income countries.** Note: Point estimates and 95% confidence intervals. N = 108 countries. All models are controlled for GDP per capita (logged), Universal Health Coverage Index, HIV Prevalence.

interventions aimed at promoting condom use and reduce sexual partners necessitating structural interventions to rectify [55]. Likewise, treatment roll-out in LMICs has been hobbled by rampant food insecurity again revealing the necessity of addressing underlying social conditions for biomedical and behavioral interventions to be efficacious [35].

While quite a few studies have attempted to articulate lessons that can be transferred from national experiences managing HIV to COVID-19, this study makes a contribution by empirically examining to what extent countries with stronger HIV policy responses also had stronger COVID-19 policy responses. However, this study has quite a few limitations that deserve further attention and qualifications on the findings. First, although we controlled for GDP per capita, UHC level, and HIV prevalence, there may be various alternative factors including international/donor pressure, regime type, the ideological character of leadership and elected representative, the overall state capacity and bureaucratic strength beyond infectious disease capacity, characteristics of the public, etc. [43, 56]. The exact mechanisms driving the associations also deserve further attention. Future research could explore case studies of countries to examine exactly how policy learning from HIV may or may not have occurred. For instance, in the US context, it is quite clear that key leadership in the COVID-19 response (including the COVID-19 task force) was drawn from the ranks of researchers and public health leaders trained in HIV and related emerging infectious diseases, providing strong experience for a cadre of researchers to apply to COVID-19. Researchers have also considered how countries recent experiences with similar outbreaks, especially SARs and MERs in Korea, but also Ebola in West Africa built public health capacity that could be more quickly mobilized and

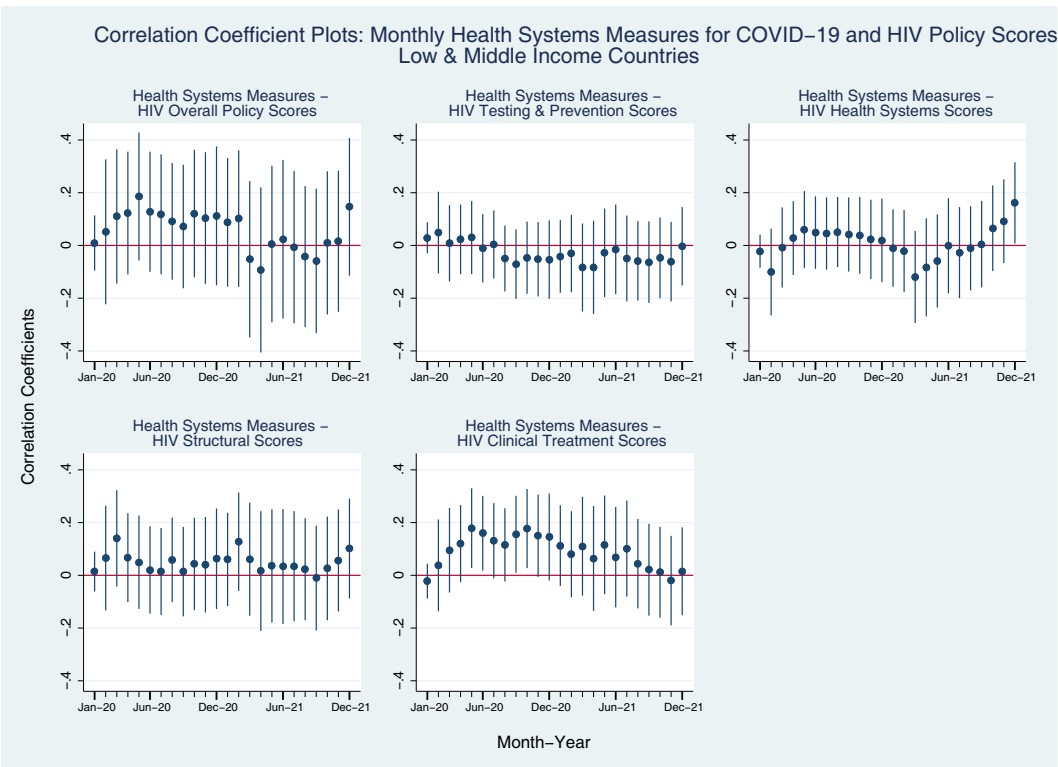

**Fig 6. Coefficient plots: Regression coefficients between COVID-19 health systems index (Y axis) and HIV policy scores (X axis) in each month from Jan 2020 to Dec 2021, low- and middle-income countries.** Note: Point estimates and 95% confidence intervals. N = 107 countries. All models are controlled for GDP per capita (logged), Universal Health Coverage Index, HIV Prevalence.

transferred to COVID-19 [57, 58]. This research also avoids thorny questions about what were (are) most "appropriate" and "effective" COVID-19 responses making it difficult to judge whether experience with HIV aided countries in making "better" COVID-19 policy decisions. Future research will have to further evaluate this question.

Finally, we must recognize the differences between COVID-19 and HIV that limits policy learning. The risk groups affected are quite different with COVID-19 morbidity affecting a broader set of less marginalized risk groups, including older adults as well as individuals with chronic conditions. The mortality burden of COVID-19 has also ultimately fallen heavier on HICs for reasons not fully understood, which distinguishes it from HIV. The higher reproductive rate of COVID-19 also serves as greater justification for enacting rights-constraining measures to preserve hospital capacity prior to the development of effective vaccines compared with HIV where quarantine measures have been deemed inappropriate violations of human rights and dignity. Countries with extensive HIV experience may have been less prepared for how to enact such broad-based restrictive measures in response to COVID-19. This is where recent experience of Ebola or SARS/MERs may have been better prepared countries compared with HIV in terms of how enforce cordon-sanitaires and gain citizen compliance.

## Conclusions

Countries with a more robust HIV policy environment responded in a more robust manner to COVID-19, particularly in regards to their economic policy response, but not necessarily in a more stringent way. Future studies should more closely examine the potential mechanisms

linking HIV policy effort and COVID-19 responses through case studies and better classify COVID-19 policy responses according to both their effectiveness and appropriateness.

## Supporting information

**S1 Table. Top and bottom 10 countries COVID-19 & HIV policy response scores.**
(DOCX)

**S2 Table. Rank of average index scores.**
(DOCX)

**S3 Table. Descriptive statistics for low- and middle-income countries.**
(DOCX)

**S1 Fig. Relationship of containment measures and HIV policy indices in first 30 days of pandemic.**
(EPS)

**S2 Fig. Relationship of economic support measures with HIV policy indices in first 30 days of pandemic.**
(EPS)

**S3 Fig. Relationship of health systems measures with HIV policy indices in first 30 days of pandemic.**
(EPS)

## Author Contributions

**Conceptualization:** Ashley Fox.

**Data curation:** Ashley Fox, Heeun Kim.

**Formal analysis:** Heeun Kim.

**Investigation:** Ashley Fox.

**Methodology:** Ashley Fox, Heeun Kim.

**Project administration:** Ashley Fox.

**Visualization:** Heeun Kim.

**Writing – original draft:** Ashley Fox, Heeun Kim.

**Writing – review & editing:** Ashley Fox, Heeun Kim.

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
