## [Decision Letter · Decision Letter 0]

26 Dec 2022

PGPH-D-22-01546

HIV Policy Legacies, Pandemic Preparedness and Policy Effort to Address COVID-19

Dear Dr. Fox,

Thank you for submitting your manuscript to PLOS Global Public Health. After careful consideration, we feel that it has merit but does not fully meet PLOS Global Public Health’s publication criteria as it currently stands. Therefore, we invite you to submit a revised version of the manuscript that addresses the points raised during the review process.

The comments from the reviewers concerning clarity of the research question; justification for how HIV policies can be useful for understanding COVID-19 response efforts (and alignment/areas of non-alignment between the diseases from a control standpoint); suggestions regarding additional justification of the methodology; and revision of how the results are presented to improve clarity are required for resubmission. The authors should also consider adjusting the methodology to change the coding of missing policies to a zero, as recommended by one reviewer, as this could represent an important source of negative data, rather than absence of data. Finally, given the highly dynamic global situation with respect to COVID-19, while it is not necessary to fully update the research by adding substantial numbers of new policies, the authors will need to update their Discussion in light of the rapidly changing national and global COVID-19 policies, and reflect on whether this impacts the generalizability of their findings. For the remaining of the comments from the peer-reviewers, please consider each of them and either revise the manuscript or provide a rebuttal accordingly. 

We look forward to receiving your revised manuscript.

Kind regards,

Claire J Standley

Academic Editor

Journal Requirements:

1. In the online submission form, you indicated that your data will be submitted to a repository upon acceptance.  We strongly recommend all authors deposit their data before acceptance, as the process can be lengthy and hold up publication timelines. Please note that, though access restrictions are acceptable now, your entire data will need to be made freely accessible if your manuscript is accepted for publication. This policy applies to all data except where public deposition would breach compliance with the protocol approved by your research ethics board. If you are unable to adhere to our open data policy, please kindly revise your statement to explain your reasoning and we will seek the editor's input on an exemption. Please be assured that, once you have provided your new statement, the assessment of your exemption will not hold up the peer review process.

Additional Editor Comments (if provided):

Thank you for the submission. The piece addresses an important topic, and provides useful data concerning potential linkages between past pandemic policies and responses to the COVID-19 pandemic. However, in its current form, the manuscript has some substantial shortcomings that would need to be addressed before further consideration. Notably, the premise of why HIV policies could provide useful information on COVID-19 response efforts, given all the differences between the diseases and current events, will be important to expand upon. Given the duration and dynamic nature of the COVID-19 pandemic, the challenge of how policies have changed over time, during different epidemiological phases (which in turn vary by country) should also be addressed directly in the manuscript - if not via updating of the methods or inclusion of new policies (i.e. into 2022) then at least as a limitation to the study. Perhaps the most extreme example of this will be the case of China, whose COVID-19 policies have changed substantially in just the past few weeks, and which could not have been anticipated by this manuscript, but yet presumably has important repercussions on the study's findings. Finally, the research has implications for how countries/authorities can better take on lessons learned, from a policy standpoint, from past epidemic experiences, for improved future pandemic preparedness and response; it would be helpful to expand upon these repercussions and opportunities for recommendations in the Discussion section.

Please also see the detailed feedback from both reviewers, provided below, in revising your manuscript for resubmission.

Reviewers' comments:

Reviewer's Responses to Questions

**Comments to the Author**

1. Does this manuscript meet PLOS Global Public Health’s publication criteria? Is the manuscript technically sound, and do the data support the conclusions? The manuscript must describe methodologically and ethically rigorous research with conclusions that are appropriately drawn based on the data presented.

Reviewer #1: Yes

Reviewer #2: Yes

2. Has the statistical analysis been performed appropriately and rigorously?

Reviewer #1: Yes

Reviewer #2: Yes

3. Have the authors made all data underlying the findings in their manuscript fully available (please refer to the Data Availability Statement at the start of the manuscript PDF file)?

Reviewer #1: Yes

Reviewer #2: Yes

4. Is the manuscript presented in an intelligible fashion and written in standard English?

Reviewer #1: No

Reviewer #2: Yes

5. Review Comments to the Author

Reviewer #1: This is a useful question to research. However, I found the article difficult to follow and the import of the findings are not entirely clear nor adequately explained. The authors should be clearer about what is in each data set and do more to clearly distinguish the Ec Supp data from Containment and Closure data. Throughout the piece it would be helpful to remind readers of the distinction. I would more clearly acknowledge the differences between HIV and COVID; for example the fact that HIV often affects marginalized and stigmatized groups and is passed sexually makes it different from COVID in very important ways. More discussion is needed about the findings and the possible reasons for the findings.

Reviewer #2: This paper examines correlations between patterns in HIV policies and COVID-19 policies to determine whether countries with strong HIV policies were better prepared to handle the COVID-19 pandemic.

This is an important question that gets at some of the underlying reasons that drive how different countries addressed the COVID-19 pandemic and provides lessons for future pandemic preparedness. However, the I think this study should do more to examine some of the nuances of this research question and provide stronger conclusions and policy guidance as a result.

Defining the research questions

The study should do more to justify the research question and explain why we might expect there to be links between HIV policy for a steady-state epidemic and a novel public health emergency.

Is the type of public health capacity needed to manage HIV in the 2017-2020 (when it is endemic and largely managed via the primary health care system) similar to what is needed to manage COVID-19 in the first two years of the pandemic? For example, do they require the same type of health infrastructure or not? For example, COVID is more dependent on acute care capacity whereas HIV relies primarily on outpatient.

Identifying the specific areas of overlap between HIV and COVID policy would strengthen the hypotheses and provide a clearer foundation for interpreting the results.

It may also be that policies mature between the first two years of an epidemic and 30-plus years into one. How relevant would we expect long-term HIV policies to match emergency-state COVID-19 policies?

The hypotheses could use some more justification so the reader understands clearly why the connections were made. For example, 30 days seems like a short time for “nuanced” policy to develop – most countries were still in crisis mode 60 or more days into the pandemic.

Methodology

For the methodology, missing data was ignored. But I would be concerned that the lack of a policy in a particular area should be taken into consideration. No data on a policy likely means the policy didn’t exist and perhaps it should be coded as a zero rather than missing. I would want to see how that change in methodology changed the results.

It would be helpful to have more justification for the methodological choices made in the study. For example, I didn’t understand why “An extra half-point is subtracted from each ordinal score if the policy [was?] targeted rather than general and with a missing value contributing 0.”

If the study data is updated daily, then perhaps the study time frame could be extended into 2022. As COVID-19 policy matured and many countries attempted to enter a “steady state” on COVID-19, I’d expect to see a stronger correlation with HIV policies.

I have several recommendations related to how the main results are presented:

The scatterplots are very hard to read. Seeing the cloud of points does not convey very useful information. I recommend simplifying their design and showing only the trend curve(s). For correlations, a line of best fit and/or correlation coefficients are a much more informative format than scatterplots alone.

I don’t think Table 1 adds very much useful information. You can simply say that the top ten top and bottom only have one in common (Rwanda) and don’t need to show the list.

You mention potential confounders in the discussion session, but I think you can do more to address that limitation in the main results section. In general, “cross-country” regressions are not seen as particularly informative since comparing the US, for example, to Zambia isn’t something that one would do otherwise. I think it would be useful to perhaps break down the analyses by continent/region and OECD/non-OECD. The patterns that relate HIV policy to COVID policy are likely different based on country income, severity of the ongoing HIV crisis and level of infrastructure that can be reallocated to a public health emergency on short notice. It would add interesting results to the paper.

Discussion and conclusion

With some adjustments to the results section, I think it would provide a much richer basis for a more detailed discussion section. The current discussion and (very short) conclusion sections leave the reader wondering what the contribution of this paper is to our understanding of how HIV policies influence responses to new public health threats. I think more effort to tease out impactful results will provide a stronger foundation for discussion, and ideally would lead to some clear take-home points for the reader.

For example, the connection between having strong social support policies for HIV and strong economic support policies for COVID is not all that surprising. Countries that have support for social policies generally apply them in multiple realms. It would be interesting to know whether this correlation is stronger or weaker depending on the level of GDP, when it might be the case that resources are not easily available for COVID social policies.

I wonder whether countries with a strong biomedical infrastructure assumed it would be able to handle fallout from the pandemic and therefore didn’t invest in social support. Generally strong biomedical infrastructure is associated with resource-rich environments. But perhaps they overestimated their ability to control COVID?

Smaller points

The references to tables A1 and A2 are reversed

The paper would benefit from more focused paragraphs and a tighter outline. It felt like it rambled a little bit too much in some sections.

It may be helpful to reference the tuberculosis literature as well since the field has struggled to find a good balance between human rights and slowing disease transmission. It has a dark history.

6. PLOS authors have the option to publish the peer review history of their article (what does this mean?). If published, this will include your full peer review and any attached files.

**Do you want your identity to be public for this peer review?** For information about this choice, including consent withdrawal, please see our Privacy Policy.

Reviewer #1: No

Reviewer #2: No

---

## [Decision Letter · Decision Letter 1]

3 Apr 2023

PGPH-D-22-01546R1

HIV Policy Legacies, Pandemic Preparedness and Policy Effort to Address COVID-19

Dear Dr. Fox,

Thank you for submitting your manuscript to PLOS Global Public Health. After careful consideration, we feel that it has merit but does not fully meet PLOS Global Public Health’s publication criteria as it currently stands. Therefore, we invite you to submit a revised version of the manuscript that addresses the points raised during the review process.

Please see further comments below regarding improving the clarity and interpretability of the figures.

We look forward to receiving your revised manuscript.

Kind regards,

Claire J Standley

Academic Editor

Journal Requirements:

Additional Editor Comments (if provided):

Please ensure all figure axes, curve lines, and other elements are clearly labeled, either directly in the figure (at appropriate font size for visibility) or in the caption. Consider removing unnecessary country labels to improve figure clarity (this is just a suggestion and is not required). 

Reviewers' comments:

Reviewer's Responses to Questions

**Comments to the Author**

1. If the authors have adequately addressed your comments raised in a previous round of review and you feel that this manuscript is now acceptable for publication, you may indicate that here to bypass the “Comments to the Author” section, enter your conflict of interest statement in the “Confidential to Editor” section, and submit your "Accept" recommendation.

Reviewer #1: All comments have been addressed

Reviewer #2: All comments have been addressed

2. Does this manuscript meet PLOS Global Public Health’s publication criteria? Is the manuscript technically sound, and do the data support the conclusions? The manuscript must describe methodologically and ethically rigorous research with conclusions that are appropriately drawn based on the data presented.

Reviewer #1: Yes

Reviewer #2: Yes

3. Has the statistical analysis been performed appropriately and rigorously?

Reviewer #1: I don't know

Reviewer #2: Yes

4. Have the authors made all data underlying the findings in their manuscript fully available (please refer to the Data Availability Statement at the start of the manuscript PDF file)?

Reviewer #1: Yes

Reviewer #2: Yes

5. Is the manuscript presented in an intelligible fashion and written in standard English?

Reviewer #1: Yes

Reviewer #2: Yes

6. Review Comments to the Author

Reviewer #1: The new draft is clearer.

Reviewer #2: The authors have addressed my concerns about the manuscript.

I do recommend some revisions to the figures for clarity since the figure design makes it hard for the reader to glean the main points. Figure 1 should have clearer X-axis labels. They are impossible to read in the current version. And the best practice is to have curves for the confidence intervals so as to focus the attention on the curves and not the interval.

For the appendix table, the cloud of points obscures the main curves in the figure. The dashed grey line is impossible to see in the color version, and a black and white version would almost obscure the red curve. There also is no legend for these curves to explain the difference between the red line and the dashed grey line. While some people might be interested in trying to make out the names of countries that fall on the edges of the cloud of points, it's really hard to read at the current size. It seems far more important to make sure the main point of the figure comes through clearly than to waste space on the few readers who want to look at the outliers.

Also, eliminating the cloud of points would allow the authors to put more curves on a single set of axes to improve the comparison between time periods. It's really hard to flip back and forth to compare between time periods with so much going on in these figures.

7. PLOS authors have the option to publish the peer review history of their article (what does this mean?). If published, this will include your full peer review and any attached files.

**Do you want your identity to be public for this peer review?** For information about this choice, including consent withdrawal, please see our Privacy Policy.

Reviewer #1: No

Reviewer #2: No

---

## [Editor Report · Decision Letter 2]

24 May 2023

HIV Policy Legacies, Pandemic Preparedness and Policy Effort to Address COVID-19

PGPH-D-22-01546R2

Dear Dr. Fox,

We are pleased to inform you that your manuscript 'HIV Policy Legacies, Pandemic Preparedness and Policy Effort to Address COVID-19' has been provisionally accepted for publication in PLOS Global Public Health.

Best regards,

Claire J Standley

Academic Editor